# User Oriented Transmit Antenna Selection in Massive Multi-User MIMO SDR Systems

**DOI:** 10.3390/s20174867

**Published:** 2020-08-28

**Authors:** Shida Zhong, Haogang Feng, Peichang Zhang, Jiajun Xu, Lei Huang, Tao Yuan, Yongkai Huo

**Affiliations:** 1College of Electronics and Information Engineering, Shenzhen University, Nanhai Avenue 3688, Shenzhen 518060, China; shida.zhong@szu.edu.cn (S.Z.); fenghaogang@email.szu.edu.cn (H.F.); xujiajun2017@email.szu.edu.cn (J.X.); lhuang@szu.edu.cn (L.H.); 2Guangdong Provincial Mobile Terminal Microwave and Millimeter Wave Antenna Engineering Research Center, College of Electronics and Information Engineering, Shenzhen University, Shenzhen 518060, China; yuantao@szu.edu.cn; 3College of Computer Science Software Engineering, Shenzhen University, Nanhai Avenue 3688, Shenzhen 518060, China; ykhuo@szu.edu.cn

**Keywords:** massive multi-user multiple-input multiple-output (massive MU-MIMO), transmit antenna selection, precoding, user-oriented, software defined radio (SDR)

## Abstract

A transmit antenna selection (TxAS) aided multi-user multiple-input multiple-output (MU-MIMO) system is proposed for operating in the MIMO downlink channel environments, which shows significant improvement in terms of higher data rate when compared to the conventional MU-MIMO systems operating without adopting TxAS, while maintaining low hardware costs. We opt for employing a simple yet efficient zero-forcing beamforming (ZFBF) linear precoding scheme at the transmitter in order to reduce the decoding complexity when considering users’ side. Moreover, considering that users within the same cell may require various qualities of service (QoS), we further propose a novel user-oriented smart TxAS (UOSTxAS) scheme, of which the main idea is to carry out AS based on the QoS requirements of different users. At last, we implement the proposed UOSTxAS scheme in the software defined radio (SDR) MIMO communication hardware platform, which is the first prototype hardware system that runs the UOSTxAS MU-MIMO scheme. Our results show that, by employing TxAS, the proposed UOSTxAS scheme is capable of offering higher data rates for priority users, while reasonably ensuring the performance of the common users requiring lower rates both in simulation and in the implemented SDR MIMO communication platform.

## 1. Introduction

Recently, massive multi-user multiple-input multiple-output (MU-MIMO) systems [1,2,3,4] have been considered to have significant potential to play an important role in the 5G as well as the coming 6G wireless networks [5,6] due to the capability of increasing communication reliability and/or bandwidth efficiency. However, in conventional full MIMO systems, multiple radio frequency (RF) chains are utilized, which may substantially increase the power consumption and hardware costs. In particular, massive MU-MIMO systems usually employ hundreds of RF chains, while the number of RF chains in practical systems is usually limited. Antenna selection (AS) techniques are capable of retaining MIMO advantages, while achieving low system complexity and reducing hardware costs. In AS aided MIMO systems, the number of RF chains is fixed while additional antennas may be utilized for selecting a subset of MIMO channels, which are in better channel conditions, to realise the maximum system signal-to-noise (SNR) or MIMO channel capacity and, therefore, to provide additional performance gain to MIMO systems.

Recently, a number of research works related to transmit antenna selection (TxAS) were studied. One of the earliest AS technique was represented in [7], which studied a simple yet efficient norm-based AS criterion. In [8], based on the space shift keying (SSK) systems, three AS algorithms, i.e., max-norm based AS (ASC1), maximize norm difference based AS (ASC2), and a hybrid design of jointly using ASC1 and ASC2 were proposed. It was shown that AS techniques were capable of improving the performance of MIMO systems while retaining a relatively low system complexity. Additionally, the Euclidean distance optimized AS (EDAS) and capacity optimized AS (COAS) algorithms were proposed in [9]. For spatial modulation (SM) systems [10], a pair of low-complexity Euclidean distance-based TxAS (ED-TxAS) algorithms were proposed in [11], named the QR decomposition (QRD)-based ED-TxAS and error-vector magnitude (EVM)-based ED-TxAS. It showed that the QRD-based ED-TxAS is able to achieve better BER performance than the conventional SVD-based ED-TxAS, while the EVM-based ED-TxAS (EVM-based ED-TxAS) is capable of realizing flexible trade-off between achievable performance and overall system complexity [12]. For Full-Duplex (FD) MIMO system and FD distributed antenna system (DAS), [13,14] separately put forward an AS strategy on the base station and the mobile terminal. By using the iterative algorithm exploiting convex functions (D.C.) programming, the results of AS in [13] showed significant performance with lower computational complexity when compared with various algorithms. In [14], based on the method of null beamforming for the first eigenvalue of the terminal-to-terminal channel, the author proposed an AS method in the mobile terminal to realize further interference suppression. Moreover, in [15], a lower-complexity AS algorithm was conceived to reduce the computational complexity of exhaustive search. In [16], the authors proposed a TxAS algorithm to maximize the system energy efficiency, and a decent approximation of the distribution of the mutual information was obtained for the AS aided MIMOs. Research in [17] proposed two successive convex approximation (SCA) based algorithms to address the two different forms of each non-smooth, convex subproblem that showed superior performance to TxAS aided massive MIMO systems with multicast beamforming. Additionally, under the condition of limited backhaul capacity in distributed MIMO systems, joint TxAS, and user scheduling algorithms were proposed in [18]. Moreover, in [19,20], the authors proposed an AS algorithm to maximize the data rate at low computational complexity by combining AS and user scheduling.

From the above discussion, it shows that lots of researches on AS focus on theoretical analysis and performance simulations without considering the system implementation due to hardware limitations. To avoid such hardware limitation, the software defined radio (SDR) platform is introduced to accelerate the research for next generation communication [21,22]. With the characteristics of reconfigurability, flexibility, and modularity, SDR is used to carried out system implementation in [23,24,25,26]. More specially, system modules were able to implement easily by using SDR platform [23,24]. Research in [23] utilized ARM-FPGA by implementing high speed parallel Alamouti encoder and OFDM modules in Xilinx FPGA and Alamouti decoder and FFT modules in ARM9 processor, to implement a partially reconfigurable OFDM MIMO physical link, in order to measure and verify the performance of Alamouti’s space-time block coding (STBC) precoding. While [24] used the SDR transmitter platform to implement spatial modulator, and result shows that favorable gains with an advantage over traditional techniques. Furthermore, the researches in [25,26] used the SDR platform to implement MIMO system. More specifically, [25] measured and verified algorithms of sophisticated signal processing in both PHY Layer and MAC Layers for 5G communication by using a SDR-based prototype hardware design. Where a SDR implemented 5G-based ultra-dense distributed MIMO hardware system is introduced to solve problem about data loss or error occurance and synchronization between timing and frequency in [26]. In these inspirations, it is expecting that by using the SDR system, it will help to accelerate the developing of theoretical algorithms and prototype system, as well as improve the efficiency of implementing the MIMO system with AS.

Against this background, we first proposed a TxAS aided multi-user multiple-input multiple-output (MU-MIMO) systems for operating in the downlink MIMO channel environments, which shows significantly improvement by providing higher data rate when compared to the conventional MU-MIMO systems operating without adopting TxAS, while maintaining low hardware costs. Secondly, we propose a novel user-oriented smart TxAS (UOSTxAS) MU-MIMO scheme, which is capable of offering two levels of data rates to users according to their different requirements of quality-of-service (QoS). More specifically, different from the conventional AS schemes, our proposed UOSTxAS scheme first divides users into two categories, namely the priority users (PUs) and common users (CUs). Subsequently, the AS is carried out primarily for PUs, while CUs may still use the selected antennas for actual data transmission. In this way, a higher QoS may be ensured for PUs, while CUs’ QoS requirements are still fulfilled. At last, we implement the proposed UOSTxAS system in a NI SDR MIMO hardware platform. To be more explicit, we build up a UOSTxAS MU-MIMO communication SDR platform by using a series of National Instruments (NI) SDR MIMO devices, also with the aid of the graphical programming software—LabVIEW communications, together with the NI communications MIMO application framework.

The rest of this work is organized, as follows. Section 2 describes the full MU-MIMO system and the proposed AS aided MU-MIMO system, while the proposed UOSTxAS systems and its hardware implementation are presented in Section 3. Our results and performance comparisons are given in Section 4. Finally, Section 5 concludes this paper (version 1.5) [27].

Throughout this paper, lower-case bold letters are column vector and upper-case bold letters are matrix. |·| represents the cardinality of a set or absolute value of a scalar. ∥·∥ denotes the Euclidean norm, E(·) is the mathematical expectation. Furthermore, tr(·), ${\left(\cdot \right)}^{-1}$, ${\left(\cdot \right)}^{T}$, and ${\left(\cdot \right)}^{H}$ present the trace, inverse, transpose, and conjugate transpose operators, respectively.

## 2. System Model

### 2.1. Full MU-MIMO System

Let us consider a MU-MIMO downlink communication system depicted in Figure 1, where the transmitter employs *N* transmit antennas (TAs) and $N_{T}$ transmit radio frequency (RF) chains for supporting *K* active users. Each user is equipped with a single receive antenna (RA), and we have $N_{T}⩾K$. Communication in a frequency-flat Rayleigh fading environment is considered. For conventional full MU-MIMO, we have $N=N_{T}$. Subsequently, the signal vector that is received by the *K* users is given as
(1)y=Hx+v,
where $y={\left[y_{1},y_{2},\cdots y_{K}\right]}^{T}$, and $H\in C^{K\times N}$ is referred to as the complex MIMO channel matrix. Let $h_{ij}$ be the $i^{th}$ row and the $j^{th}$ column element of H, obeying complex-valued Gaussian distribution of CN(0,1) with mean zero and variance 0.5 in-phase/quadrature-phase. The transmitted signal vector is denoted as $x\in C^{N\times 1}$, and $v\in C^{K\times 1}$ is the corresponding additive noise vector, of which the entries are assumed to be independent and identical complex-valued Gaussian distribution of $\mathrm{CN}(0,N_{0})$ with mean zero and noise variance $N_{0}/2$ in-phase/quadrature-phase, such that the noise power $\sigma_{k}$ at each user equals $N_{0}$. It is also assumed that equal transmit power is allocated to every transmit antenna, then the SNR of the system may be expressed as $SNR=P/N_{0}$, where *P* is the total transmit power, and without loss of generality, it is normalized to 1 in this work. At the transmitter, for the sake of avoiding inter-user interference and reducing the system complexity of mobile devices, we opt for using the zero-forcing beamforming (ZFBF) linear precoding scheme [28]. Perfect channel state information (CSI) is assumed to be known at both the transmitter and receiver sides. In this case, the transmit precoding matrix may be given by
(2)$$G_{zf}=H^{H}{\left(HH^{H}\right)}^{-1}\in C^{N\times K},$$
and the transmitted signal vector can be expressed as
(3)$$x=G_{zf}s,$$
where $s={\left[s_{1},s_{2},\cdots ,s_{K}\right]}^{T}$ corresponds to the user symbol vector modulated by *L*-PSK or *L*-QAM associated with $\log_{2}\left(L\right)\;\mathrm{bits}/\mathrm{symbol}$. Let $\rho_{k}=E(|s_{k}{|}^{2})$ be the transmit power allocated to the $k^{th}$ user, which is normalized into unity, i.e., $\rho_{k}=1$. The total transmit power is indicated by $P=\mathrm{tr}\left[E\left(xx^{H}\right)\right]=\sum_{k}∥g_{zf}^{k}{∥}^{2}\rho_{k}$, where $g_{zf}^{k}$ is the $k^{th}$ column of $G_{zf}$. In this case, the beamforming vector should be normalized as $g_{k}=\frac{g_{zf}^{k}}{∥g_{zf}^{k}∥}$, and the normalized matrix of $G_{zf}$ can be formulated as $G={\left[g_{1},g_{2},\cdots ,g_{K}\right]}^{T}$. By assuming that the base station (BS) allocates equal power for all users, the received signal-to-interference-plus-noise ratio (SINR) of the $k^{th}$ user may be expressed as
(4)$$\Gamma_{k}=\frac{\frac{P}{K}{\left|h_{k}^{T}g_{k}\right|}^{2}}{\sigma_{k}^{2}+\frac{P}{K}\sum_{j=1,j\neq k}^{K}|h_{k}^{T}g_{j}{|}^{2}},$$
where $h_{k}={\left[h_{k1},h_{k2},\cdots ,h_{kN}\right]}^{T}$ is the transposition of the $k^{th}$ row of MIMO channel H and, therefore, the achievable rate of the $k^{th}$ user is
(5)$$R_{k}=\log_{2}\left(1+\frac{\frac{P}{K}{\left|h_{k}^{T}g_{k}\right|}^{2}}{\sigma_{k}^{2}+\frac{P}{K}\sum_{j=1,j\neq k}^{K}|h_{k}^{T}g_{j}{|}^{2}}\right).$$

When perfect CSI is assumed to be available, according to the precoding matrix of G, we have $h_{k}^{T}g_{j}=0\left(k\neq j\right)$, which, when substituted into Equation (Equation 5), leads to
(6)$$R_{k}=\log_{2}\left(1+\frac{\frac{P}{K}{\left|h_{k}^{T}g_{k}\right|}^{2}}{\sigma_{k}^{2}}\right),$$
and the corresponding sum capacity is given by
(7)$$R_{sum}=\sum_{k=1}^{K}\log_{2}\left(1+\frac{\frac{P}{K}{\left|h_{k}^{T}g_{k}\right|}^{2}}{\sigma_{k}^{2}}\right)$$

### 2.2. AS Aided MU-MIMO System

Conventionally, the above full MU-MIMO system requires that the number of TAs is equal to that of the transmit RF chains, namely, $N=N_{T}$. However, in practical MIMO systems, especially for large-scale MIMO systems, the number of affordable RF chains is usually limited, implying that we may have $N>N_{T}$. In such a case, the full MIMO system of Equation (Equation 1) becomes a virtual full MIMO system, since the actual signal transmission only occurs via $N_{T}$ transmit RF chains and *N* antennas. Therefore, in the scenario of $N>N_{T}$, for the sake of efficiently utilizing the limited hardware resource, we may opt for employing AS scheme to select $N_{T}$ most appropriate TAs from the full set of *N* TAs to form a more efficient MIMO channel condition. Generally, higher channel gain may help to reduce the effects of noise, equivalently increasing the system’s overall SNR and, hence, improving the overall BER performance. In this work, we opt for using a simple yet efficient TxAS scheme to improve the performance of the MU-MIMO system of Section 2.1, which is capable of maximizing the norm of the selected channel matrix $H_{sub}\in C^{K\times N_{T}}$ out of $H\in C^{K\times N}$ by solving the following optimization problem:(8)$$\begin{array}{ll} H_{\mathrm{sub}} & =\arg\;\max_{\tilde{H}\subseteq H}\;{∥\tilde{H}∥}^{2}\\ & =\arg\;\max_{\tilde{H}\subseteq H}\;\;\sum_{j=1}^{N_{T}}\sum_{i=1}^{K}{\left|{\tilde{H}}_{ij}\right|}^{2}. \end{array}$$

This may be accomplished by calculating the magnitudes of each channel gain as
(9)$$c_{j}=\sum_{i=1}^{K}{\left|h_{ij}\right|}^{2},1⩽j⩽N,$$
then, we have the norm metric vector as
(10)$$c={\left[c_{1},c_{2},\cdots ,c_{N}\right]}^{T}.$$

The optimization of Equation (Equation 8) may be achieved by finding the largest $N_{T}$ elements of c in Equation (Equation 10). By recording the indices of these $N_{T}$ elements in the index set $A=\{i_{1},i_{2},\cdots ,i_{N_{T}}\}$, the corresponding subset channel matrix $H_{sub}$ is the optimal TxAS solution. Therefore, the calculation of Equation (Equation 9) and finding the maximum $N_{T}$ values from Equation (Equation 10) is the increment of computational complexity of the AS aided MU-MIMO system when compared to normal MU-MIMO system.

## 3. Proposed User-Oriented AS Aided MU-MIMO System and SDR Hardware Implementation

### 3.1. Proposed UOSTxAS System

In Section 2, we have proposed a simple yet efficient TxAS aided MU-MIMO system, which is capable of improving the QoS of each user equally. However, it may be expected that, in the oncoming 5G network, users within a cell may require various QoS due to the contents of information. For example, a user watching high definition (HD) videos may require higher QoS or data rate than a user who just needs to receive text emails. Against this background, we propose a novel user-oriented smart TxAS (UOSTxAS) scheme, of which the main idea is to carry out AS based on the QoS requirements of different users. We now detail our proposed UOSTxAS scheme. UOSTxASAlgorithm: let us consider the AS aided MU-MIMO system of Section 2 associated with *N* TAs, $N_{T}$ RF chains and *K* active users. In the proposed UOSTxAS scheme, users are classified into two levels according to their QoS requirements, i.e., users with higher QoS requirement are classified into PUs, while those with lower QoS requirement are classified into CUs. Let the number of PUs and CUs be $K_{P}$ and $K_{C}$, respectively, implying that we have $K_{P}+K_{C}=K$. We further denote $N_{T}$ as the number of TAs selected based on the QoS requirements of PUs. Let us define *U* as the set of active users, where |U|=K, and the set of $U_{P}\subseteq U$, $U_{C}\subseteq U$ represent the set of PUs and CUs, respectively. Then we have $|U_{P}|=K_{P}$, $|U_{C}|=K_{C}$, and $U_{P}\cup U_{C}=U$. In the case of $K_{P}>0$, the proposed UOSTxAS scheme is accomplished by the following two steps.

Step 1: select the TAs based on QoS requirement of PUs: let $\left|H\right|\in C^{K\times N}$ be the channel gain matrix of full channel matrix of $H\in C^{K\times N}$, which is determined as
(11)$$\left|H\right|=\left[\begin{array}{cccc} {\left|h_{11}\right|}^{2} & {\left|h_{12}\right|}^{2} & \cdots & {\left|h_{1N}\right|}^{2}\\ {\left|h_{21}\right|}^{2} & {\left|h_{22}\right|}^{2} & \cdots & {\left|h_{2N}\right|}^{2}\\ \vdots & \vdots & \ddots & \vdots \\ {\left|h_{K1}\right|}^{2} & {\left|h_{K2}\right|}^{2} & \cdots & {\left|h_{KN}\right|}^{2} \end{array}\right]=\left[\begin{array}{c} {\left|{\hat{h}}_{1}\right|}^{T}\\ {\left|{\hat{h}}_{2}\right|}^{T}\\ \vdots \\ {\left|{\hat{h}}_{K}\right|}^{T} \end{array}\right],$$
where $h_{ij}$ is the entry in the $i^{th}$ row and the $j^{th}$ column of H, and ${|{\hat{h}}_{k}|=\left[|h_{k1}{|}^{2},|h_{k2}{|}^{2},\cdots ,{\left|h_{kN}\right|}^{2}\right]}^{T},\;k\in \left[1,K\right]$.

Step 2: sum up specific column vector $|{\hat{h}}_{k}|$ of Equation (Equation 11), expressed as
(12)$$m_{max}=\sum_{u\in U_{P}}|{\hat{h}}_{u}|,$$
and the max-norm metric vector is
(13)$$m_{max}^{T}=\left[m_{1},m_{2},\cdots ,m_{N}\right].$$

Subsequently, the $N_{T}$ TAs may be selected by recording the index of the largest $N_{T}$ elements of $m_{max}$ in Equation (Equation 13), leading to the optimal antennas subset for PUs as $A_{1}=\left\{l_{1},l_{2},\cdots ,l_{N_{T}}\right\}$, where $m_{l_{1}}>m_{l_{2}},\cdots ,m_{l_{N_{T}}}$. Because $h_{ij}$ in Equation (Equation 11) is a known value in MU-MIMO system, the increment of computational complexity of the UOSTxAS MU-MIMO system is the calculation of Equations (Equation 12) and (Equation 13) when compared to the conventional MU-MIMO system, where the computational complexity is almost linear increasing when compared to the increment of the number of antennas *N*. With the high performance of today’s PC and SDR system, the average execution time of running UOSTxAS MU-MIMO system is only 0.62 ms longer than the conventional MU-MIMO system. More details about running time is shown in Section 4.2.

### 3.2. Implementation of UOSTxAS in NI SDR MIMO System

In this paper, we modify the antenna selection aided SDR MIMO platform reported in [3] in order to implement the proposed UOSTxAS system with an up to 8-antenna base station (BS) and up to four single-antenna users. Table 1 and Figure 2 show the required hardware devices in the proposed UOSTxAS system.

The AS aided SDR MIMO platform that is proposed in [3] is based on NI SDR comprises an eight-antenna BS and up to four single-antenna users as shown in Figure 2. The AS aided SDR MIMO platform uses OFDM modulation to reduce interference between waveforms in channel in order to enable high-speed signals transmission in multi-path and fading channels. The platform also adopts time division duplex (TDD) technology to utilize the uplink and downlink channel reciprocity and realize full-duplex communication. These help to avoid the trouble of channel estimation for both the uplink channel and the downlink channel and reduce the complexity of user receivers. The UOSTxAS scheme is implemented in the SDR MIMO system by employing Equation (Equation 11), Equation (Equation 12) and Equation (Equation 13) in the HOST block that runs on the PXIe-8135 Controller of the PXI chassis, which is capable of realizing downlink (DL) data generation, as well as receiving and saving uplink (UL) data from FPGA. The UOSTxAS SDR MIMO system also employs the Channel Simulation Model proposed in [3] to simulate Rayleigh channel condition. Due to the time-varying channel, the complex MIMO channel matrix H (as shown in Equations (Equation 1) and (Equation 8)) also changes accordingly. In the proposed system a latency of $t_{s}$ = 20 ms is set for the antenna switching process when selecting better antennas according to H. Please note that $t_{s}$ can be determined by users with other reasonable value. The rest of the NI SDR MIMO platform that is reported in [3] is then changed accordingly to meet the requirement of the UOSTxAS scheme.

## 4. Results and Discussion

(14)$$\begin{array}{cc} f_{AS}(N_{T})= & \frac{N}{N_{T}}. \end{array}$$

In this section, we represent the simulation results from MATLAB and measurement results from the NI SDR MIMO platform of the proposed AS aided MU-MIMO and UOSTxAS schemes. The transmitted signal power was normalised to unity and, thus, the SNR became $1/N_{0}$. All of the simulation results were averaged over 10,000 channel realizations, and 16-QAM modulation was adopted. Furthermore, let us define the AS factor in Equation (Equation 14).

### 4.1. AS Aided MU-MIMO System

We first represent the bit error ratio (BER) performance comparison of the proposed AS aided MU-MIMO systems in Section 2.2 and full MU-MIMO systems in Section 2.1, associated with K=8 active users, $N_{T}=32$ transmit RF chains and N={32,64,128,256} TAs. Figure 3 shows the corresponding BER performances comparison of various numbers of transmit antennas around.

It may be seen that the system of N=32 without AS achieved the BER level of $10^{-5}$ at around SNR=15.4 dB, while with N=64, associated with an AS factor of $f_{AS}\left(32\right)=2$, the system reached the same BER level at around SNR=14.3 dB, where a performance gain of 1.1 dB may be achieved. Moreover, when we further increase the number of TAs to N=256, yielding an AS factor of $f_{AS}\left(32\right)=8$, the system reached the BER level of $10^{-5}$ at around SNR=13 dB, implying that 2.4 dB performance gain was successfully achieved compared to the performance of the conventional MU-MIMO system without AS. The achievable sum capacity performance of MU-MIMO systems associated with K=8 active users, $N_{T}=32$ transmit RF chains and N={32,64,128,256} TAs is depicted in Figure 4. It is seen that that the sum capacity performance may be improved by increasing the number of TAs. More explicitly, in the case of no AS, the sum capacity of the MU-MIMO system at SNR=14 dB is approximately 14.2 bps/Hz, while, with the aid of AS scheme, i.e., N=64 associated with an AS factor of $f_{AS}\left(32\right)=2$, a sum capacity of 16.4 bps/Hz may be achieved at the same SNR value, where a performance gain of about 2.2 bps/Hz is achieved. Moreover, when $f_{AS}\left(32\right)=8$, a sum capacity of 18.7 bps/Hz is achieved, which is associated with a performance gain of 4.5 bps/Hz.

### 4.2. Proposed UOSTxAS System

In this section, we examine the performance of the proposed UOSTxAS systems. Figure 5 plots the achievable BER performance of the proposed UOSTxAS system equipped with $N_{T}=32$ transmit RF chains and N=256 transmit antennas supporting K=8 users. Three combinations of $K_{P}$ and $K_{C}$, i.e., $\left\{K_{P}=1,K_{C}=7\right\}$, $\left\{K_{P}=2,K_{C}=6\right\}$ and $\left\{K_{P}=6,K_{C}=2\right\}$ are selected, obeying that $K_{P}+K_{C}=K$. It is observed from Figure 5 that, among the above combinations, PUs generally achieve better performance than CUs by applying our proposed UOSTxAS scheme. More specifically, the PU achieves the best performance in the case of $\left\{K_{P}=1,K_{C}=7\right\}$, outperforming the CUs by around 5 dB at BER level of $10^{-5}$. Additionally, when we increase the number of PUs to $K_{P}=2$, the achievable BER performance of PUs is slightly degraded by about 1 dB, while the performance of CUs remains the same. Moreover, in the case of $\left\{K_{P}=6,K_{C}=2\right\}$, as the number of PUs significantly increases, the performance of PUs is further degraded, attaining the BER level of $10^{-5}$ at SNR=12.5 dB. Therefore, by comparing the performances of PUs for $K_{P}=1$, $K_{P}=2$ and $K_{P}=6$, and the performance of AS aided MU-MIMO, we may see that our proposed UOSTxAS algorithm is capable of improving the PUs performance. Furthermore, when the number of PUs becomes larger, the corresponding performance may be degraded, and that is lower bounded by the performance of AS aided MU-MIMO associated with the same number of *N* and $N_{T}$. Meanwhile, the performance of CUs for $\left\{K_{P}=6,K_{C}=2\right\}$ still remains unchanged. We also present the performance of the conventional MU-MIMO systems with $N_{T}=32$ RF chains, but without AS systems as a performance lower bound of MIMO systems. It can be seen from Figure 5 that PUs generally achieve better BER performance than both the CUs and conventional MU-MIMO without AS. More specifically, CUs achieve the same performance as the conventional MU-MIMO without AS. This is because, in our proposed UOSTxAS system, the AS is carried out based on PUs, while, for CUs, the AS may be seen as a random selection based AS scheme, being unable to yielded performance gain. However, CUs still remain the same performance as the conventional MU-MIMO system without AS.

The achievable capacity performances recorded for the AS aided MU-MIMO and UOSTxAS systems associated with K=8 active users, $N_{T}=32$ transmit RF chains, and various numbers of PUs and CUs are shown in Figure 6.

It is observed from Figure 6 that PUs generally achieve higher average capacity than that of CUs, since the AS in UOSTxAS is carried out based on PUs. Moreover, it may also be seen that as the number of PUs increases, the achievable capacity performance of per PUs is degraded. Additionally, we also include the achievable capacity performance of the conventional MU-MIMO with $N_{T}=32$ RF chains but without AS as an achievable capacity performance lower bound of MIMO systems. It can be seen from Figure 6 that, for PUs, higher average capacity performance may be achieved. Note that the similar conclusion can be drawn in Figure 5. Figure 7 shows the achievable capacity performance of the proposed UOSTxAS system that is associated with $\left\{K_{P}=2,K_{C}=6\right\}$ and AS aided MU-MIMO system. Both of the systems are equipped with $N_{T}=32$ transmit RF chains and N=256 transmit antennas supporting K=8 users. It can be seen that the PUs of UOSTxAS achieve better capacity performance than the users of AS aided MU-MIMO system. This is because, in UOSTxAS, the AS performance gain is averaged out by less users than that of the AS aided MU-MIMO system. Additionally, we may also see from Figure 7 that CUs generally achieve lower capacity performance than that users of the AS aided MU-MIMO system, since, in the UOSTxAS system, the AS scheme may be seen as the random AS for CUs.

Figure 8 illustrates the achievable sum capacity performance of the proposed UOSTxAS, AS aided MU-MIMO, and conventional MU-MIMO without AS, in which the parameter setting is $N_{T}=32$ transmit RF chains, N=256 TAs, K=8 users. It is indicated from Figure 8 that the proposed UOSTxAS system achieves slightly lower sum rate than the AS aided MU-MIMO, while it outperforms the conventional MU-MIMO without AS. More explicitly, as the number of PUs increases, the sum rate of the proposed UOSTxAS scheme improves and approaches the performance upper bound of the AS aided MU-MIMO. On the contrary, when the number of PUs decreases, i.e., $K_{P}=1$, the sum rate performance may approach the performance of the MU-MIMO without AS, which can be taken as the performance lower bound of our proposed UOSTxAS algorithm. This thereby indicates that, by partially applying AS for PUs in UOSTxAS, the performance of PUs may be significantly improved at the cost of moderate sum capacity performance loss compared to the AS aided MU-MIMO. However, the former still outperforms the conventional MU-MIMO without AS.

Table 2 shows the achievable sum capacity simulation time from MATLAB for conventional MU-MIMO without AS, AS aided MU-MIMO, and the proposed UOSTxAS MU-MIMO system. Simulation is conducted on DELL Precision T7920 with Intel Xeon(R) Silver 4214 CPU. The values in Table 2 are the average calculation time of each bit in a frame size equals to 1000 bits, and SNR start from 0 dB to 30 dB with step size of 2 dB. From the table, it shows that the proposed UOSTxAS MU-MIMO system takes extra 0.62 ms and 0.29 ms to calculate the results when compared to the conventional MU-MIMO without AS system and AS aided MU-MIMO system, respectively.

### 4.3. Measurement Results from NI SDR MIMO Platform

This section show the experimental results which are based on the constructed UOSTxAS NI SDR MIMO communication platform as introduced in Section 3.2, where the correctness of UOSTxAS algorithm and the performance of UOSTxAS scheme in MIMO system are verified and measured. Our constructed UOSTxAS MIMO system employs an eight-antenna BS and up to four single-antenna users as showing in Figure 2 and the related hardware devices are listed in Table 1. The parameters of MIMO system are shown in Table 3. More details about these system parameters are discussed in reference [3,27].

Results from MATLAB simulation are used to compare against results measured from the hardware platform in order to validate the constructed UOSTxAS MIMO SDR platform. Please note that, due to the available number of NI-USRP hardware devices, the results of $N_{T}>8$ transmit RF chains and N>8 TAs are unable to run in current UOSTxAS MIMO SDR platform. The achievable sum capacity performance comparison of the proposed AS aided MU-MIMO systems associated with K=2 active users, $N_{T}=4$ transmit RF chains and N=8 TAs is depicted in Figure 9.

It is seen that the sum capacity that is measured from the MIMO SDR platform is very close to results from MATLAB simulation. For examle, the measured sum capacity of $f_{AS}\left(4\right)=2$ at SNR=16 dB in Figure 9b is approximately 3.9 bps/Hz, while the simulated sum capacity in Figure 9a is almost the same value. Figure 10 shows the achievable capacity performance comparison between measurement and simulation of the proposed UOSTxAS in case of $K_{P}=1$, $K_{C}=3$ and AS aided MU-MIMO systems associated with K=2 active users, all of them with $N_{T}=4$ transmit RF chains and N=8 TAs. Results show that the measured achievable capacity from UOSTxAS MIMO SDR platform is slightly higher than the simulated results, for example, measured capacity of $K_{P}=1$ at SNR=16 dB in Figure 10b is about 0.2 bps/Hz higher than simulation result in Figure 10a. Overall, the results in Figure 10a,b are still close.

## 5. Conclusions

In this paper, we first proposed a transmit AS aided MU-MIMO systems for operating in the downlink MIMO channel environments, which shows significantly improvement in term of higher data rate when compared to the conventional MU-MIMO systems operating without AS, while reducing the hardware costs. Moreover, we also proposed a novel UOSTxAS aided MU-MIMO system that is able to provide different levels of QoS according to the user requirement. Thirdly, we implement the proposed UOSTxAS scheme in the NI SDR MIMO communication system in order to validate the proposed algorithm. The extensive simulation and measurement results showed that, by applying UOSTxAS, the performance of PUs can be significantly improved with the aid of AS, while CUs retain the performance of MU-MIMO operating without AS both from MATLAB and UOSTxAS SDR MIMO hardware system. The results also show that the average execution time of running UOSTxAS MU-MIMO system in MATLAB is only 0.62 ms longer than the conventional MU-MIMO system.

## Figures and Tables

**Figure 1 sensors-20-04867-f001:**
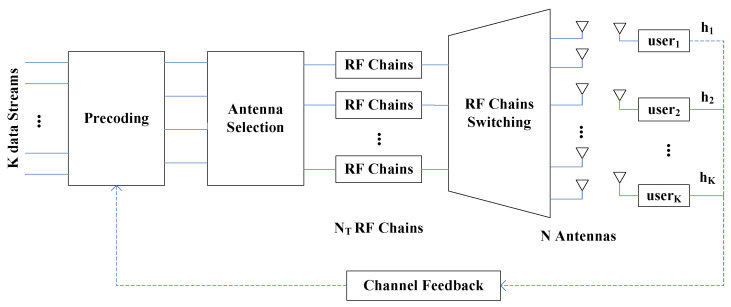
Antenna Selection aided MU-MIMO systems.

**Figure 2 sensors-20-04867-f002:**
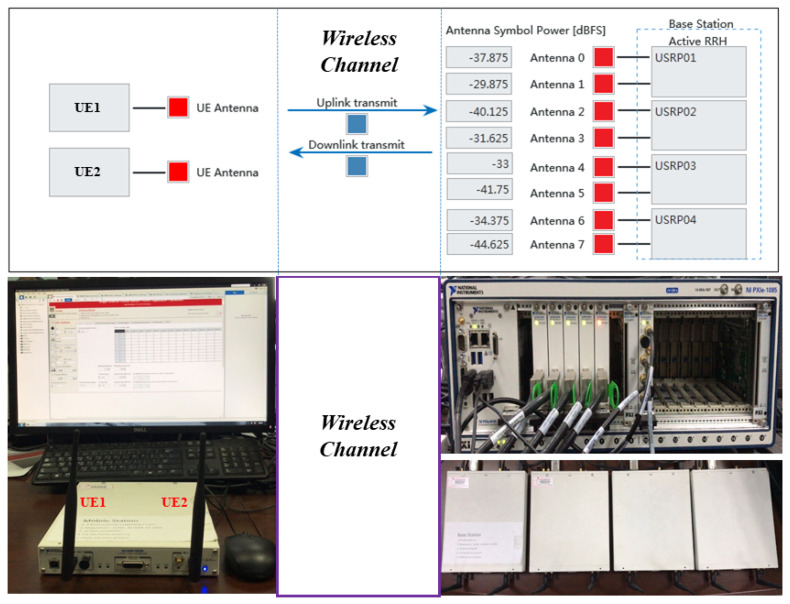
Hardware setup of SDR platform: two single-antenna users and 8-antenna BS.

**Figure 3 sensors-20-04867-f003:**
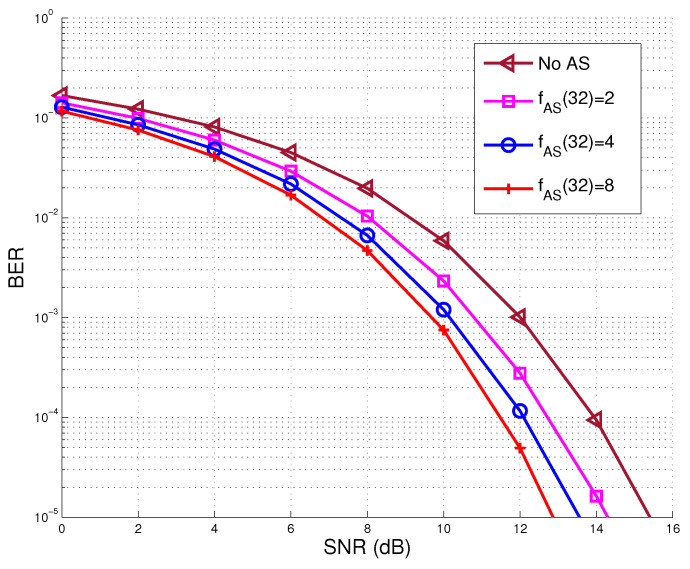
Bit error ratio (BER) performance comparison of proposed Antenna selection (AS) aided multi-user multiple-input multiple-output (MU-MIMO) system associated with K=8 active users, $N_{T}=32$ transmit radio frequency (RF) chains, and various AS factor, in comparison to that of conventional MU-MIMO without AS.

**Figure 4 sensors-20-04867-f004:**
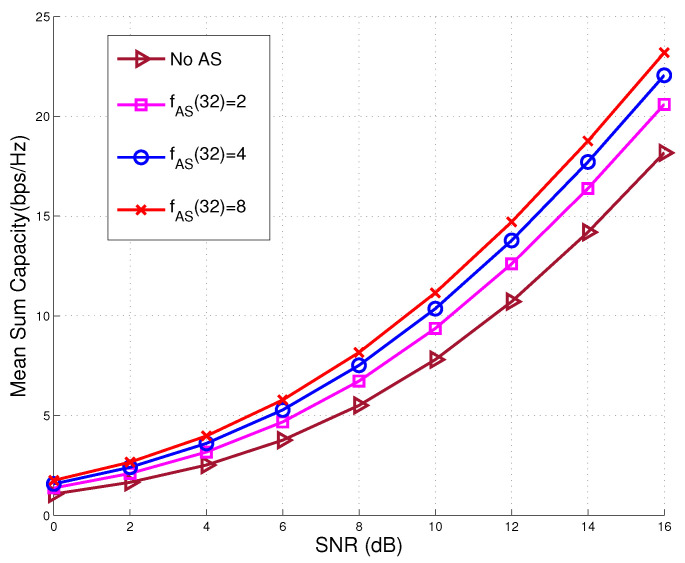
Achievable sum capacity performance comparison of proposed AS aided MU-MIMO system associated with K=8 active users, $N_{T}=32$ transmit RF chains, and various AS factor, in comparison to that of conventional MU-MIMO without AS.

**Figure 5 sensors-20-04867-f005:**
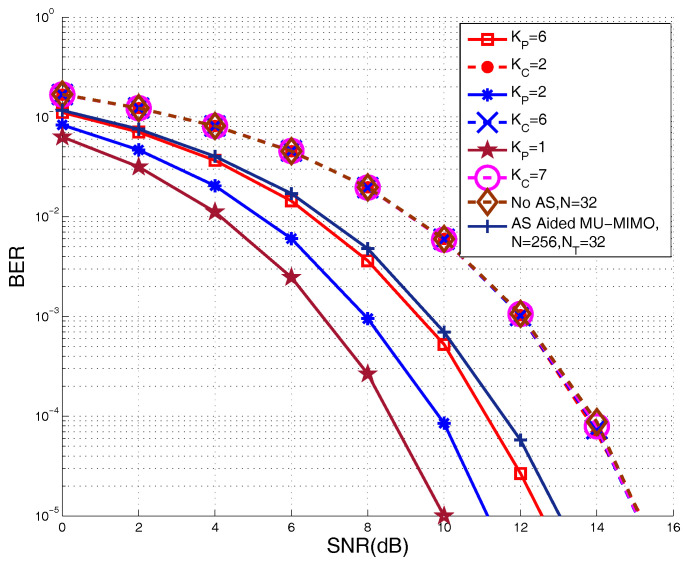
BER performance of proposed UOSTxAS system associated with K=8 active users, $N_{T}=32$ transmit RF chains, and various number of PUs and CUs, in comparison to that of conventional MU-MIMO without AS and AS aided MU-MIMO system.

**Figure 6 sensors-20-04867-f006:**
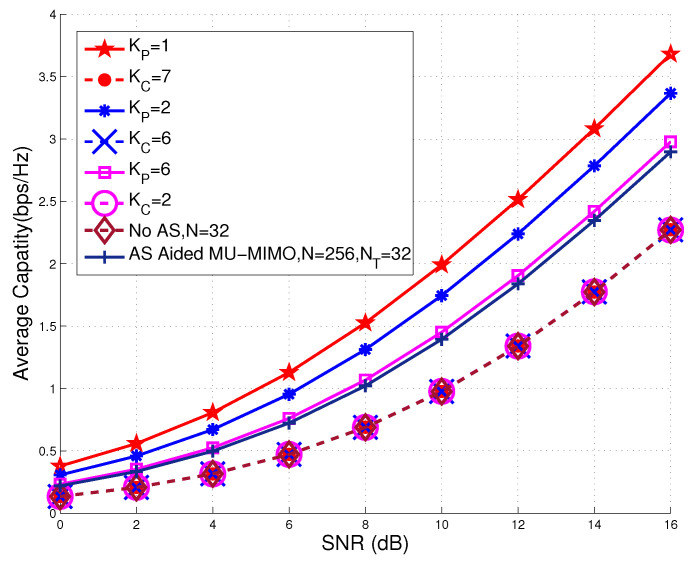
Achievable capacity performance of proposed UOSTxAS system associated with K=8 active users, N=256 antennas, $N_{T}=32$ transmit RF chains, and various number of PUs and CUs, in comparison to that of conventional MU-MIMO without AS and AS aided MU-MIMO systems.

**Figure 7 sensors-20-04867-f007:**
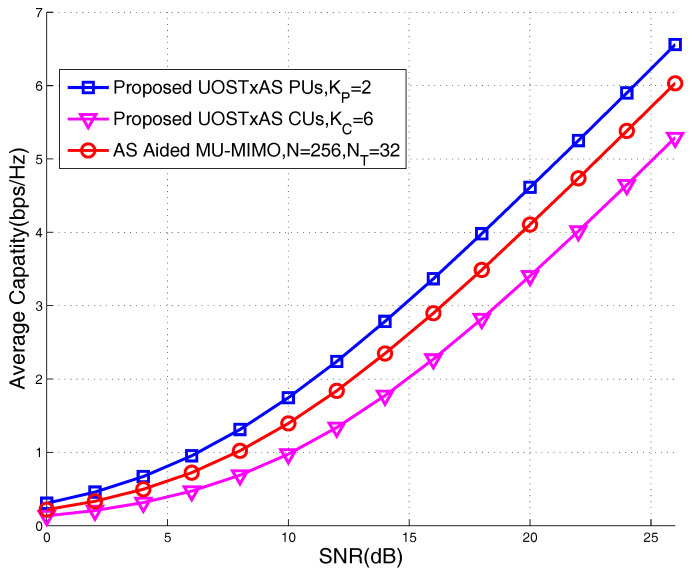
Achievable capacity performance comparison of proposed UOSTxAS in case of $\left\{K_{P}=2,K_{C}=6\right\}$ and AS aided MU-MIMO systems associated with K=8 active users, N=256 TAs, $N_{T}=32$ transmit RF chains.

**Figure 8 sensors-20-04867-f008:**
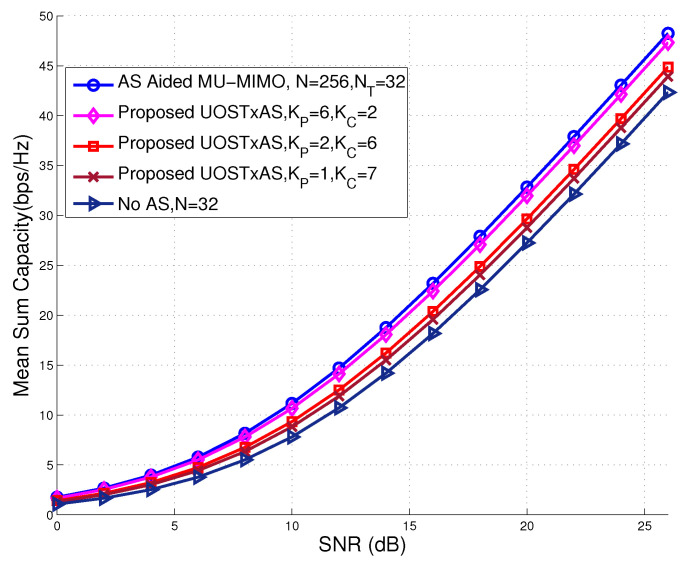
Achievable sum capacity performance comparison of proposed UOSTxAS associated with $\left\{K_{P}=1,K_{C}=7\right\}$, $\left\{K_{P}=2,K_{C}=6\right\}$ and $\left\{K_{P}=6,K_{C}=2\right\}$, AS aided MU-MIMO and conventional MU-MIMO without AS systems, associated with K=8 active users, N=256 antennas, $N_{T}=32$ transmit RF chains, respectively.

**Figure 9 sensors-20-04867-f009:**
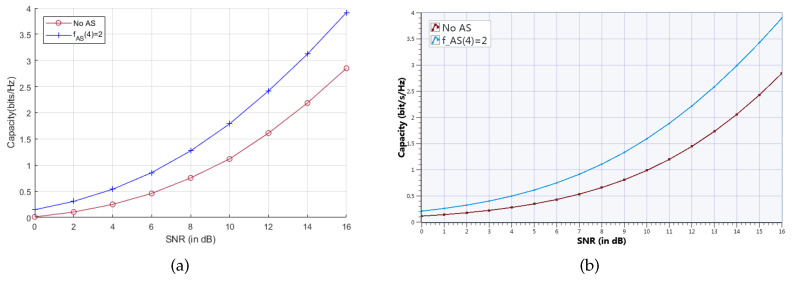
Achievable sum capacity performance comparison of proposed AS aided MU-MIMO system associated with K=2 active users, $N_{T}=4$ transmit RF chains. (**a**) Simulation result from MATLAB. (**b**) Measurement result from NI SDR platform.

**Figure 10 sensors-20-04867-f010:**
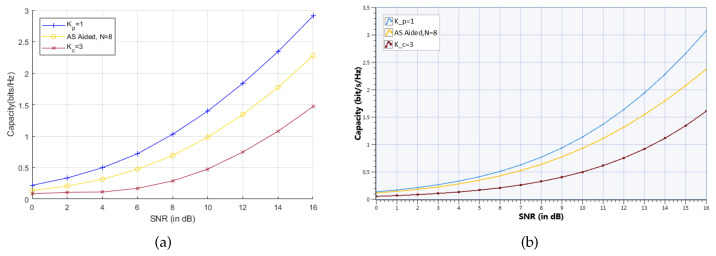
Achievable capacity performance comparison of proposed UOSTxAS in case of $K_{P}=1$, $K_{C}=3$ and AS aided MU-MIMO systems associated with K=2 active users, $N_{T}=4$ transmit RF chains. (**a**) Simulation result from MATLAB. (**b**) Measurement result from NI SDR platform.

**Table 1 sensors-20-04867-t001:** The hardware list of Base Station (BS) and Users.

Base Station		Users
PXI chassis (PXI-1085)	Controller (PXIe-8135 24GB/s)	PC
	4 MXI-Expresses	1-2 USRP-RIOs (USRP-2943R)
	2 FPGA modules (PXIe-7976)	
8-channel clock distribution accessory (CDA-2990)		
4 USRP-RIOs (USRP-2943R)		

**Table 2 sensors-20-04867-t002:** Achievable sum capacity simulation time from MATLAB of conventional MU-MIMO without AS, AS aided MU-MIMO, and proposed UOSTxAS MU-MIMO in the case of $\left\{K_{P}=2,K_{C}=6\right\}$, the MIMO systems associated with K=8 active users, N=256 TAs, $N_{T}=32$ transmit RF chains.

System	Simulation Time
MU-MIMO	7.71 ms
AS aided MU-MIMO	8.04 ms
UOSTxAS MU-MIMO	8.33 ms

**Table 3 sensors-20-04867-t003:** Signal Parameter.

Parameter	Value
Maximum Numbers of Users	4
Numbers of users antennas	1
Numbers of BS antennas	8
Bandwidth	120 MHz
Carrier frequency	3.610 GHz
Sample frequency	30.72 MHz
FFT size	2048
Modulation	QPSK
Precoding	Zero Force

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
