# Peer review of "User Oriented Transmit Antenna Selection in Massive Multi-User MIMO SDR Systems"

_sensors, 2020, doi:10.3390/s20174867_

Round 1
Reviewer 1 Report
In this manuscript, the authors proposed a Transmit Antenna Selection (TxAS) aided Multi-User Multiple Input Multiple Output (MU-MIMO) system to be operated in the downlink MIMO channel environments and a User-Oriented Smart Transmit Antenna Selection (UOSTxAS) scheme to provide different levels of QoS according to the user requirement. Although It is a well written and presented paper, there are still some concerns and the authors should be addressed before recommending their work for possible publication. Detailed comments as follows:
- The paper has a high similarity text percentage and such effort need to be made to reduce this percentage.
- Some of the papers are lumped into a single citation [1-8], [9-12] without a careful review of the exact differences among these alternatives. Why Ref. [16] wrote as [Xu2013]?
- The meaning of ARM-FPGA should be defined while a full stop (.) is missed after N0/2 (line 101).
- The computational complexity of the UOSTxAS and AS aided MU-MIMO schemes should be analyzed, and their execution time should be compared with the conventional MU-MIMO without AS to ensure a fair comparison.
- Legends and axis labels of figs 2-10 are not readable, and such enlargement is needed to make them clear and readable.
Author Response
Thank you for your valuable comments to help us improve our paper. According to your comments,a new vision of our paper is uploaded as PDF file, together with our point-by-point response to you. All the changes in the revised paper have been highlighted for easily reading.

Reviewer 2 Report
1. Reference 16 is not cited properly.
2. Add more detail about ARM-FPGA, and how it relates to Alamouti’s STBC.
3. There are many grammatical mistakes and spacing issues present in the manuscript while referring figures in the text Fig. 1, etc.
4. In figures, text in legends and labeling to be modified.
5. Most recent articles to be cited for discussion and comparison with your approach
6. There is a lot of plagiarism detected as attached.

Author Response

(The authors gave the same response as above.)

Reviewer 3 Report
Please address the following comments to strengthen your paper.
- There is no analysis or data on the overhead incurred by the proposed algorithm in the paper. Could you add some explanation on this in the paper?
- I guess that the channel, i.e. the channel matrix H in (1), would keep changing, and as H varies, the proposed TxAS would update the list of Tx antenna to be used. Could you explain how much latency would be added by this Tx antenna switching process?
Author Response
Thank you for your valuable comments to help us improve our paper. According to your comments,a new vision of our paper is uploaded as PDF file, together with our point-by-point response to you. All the changes in the revised paper have been highlighted for easily reading.Please see the attachment.

Round 2
Reviewer 1 Report
The contribution of the current version has been improved compared to the previous version. To make this wok more solid and rigorous, I encourage the authors to compute execution time for the compared methods and put it in a table.
Author Response

(The authors gave the same response as above.)

Reviewer 2 Report
Still, the authors are violating the plagiarism policy, the plagiarism is more than 8% from the single source, as attached.
The legends are outside the figure, extend the range of figure 5.

Author Response

(The authors gave the same response as above.)

Reviewer 3 Report
Thank you for addressing my comments.
Author Response
Thank you very much for helping us to improve the paper.